# Children Naturally Evading COVID-19—Why Children Differ from Adults

**Camilla Adler Sørensen, Amanda Clemmensen, Cillia Sparrewath, Malte Mose Tetens and Karen Angeliki Krogfelt ***

Department of Science and Environment, Roskilde University, 4000 Roskilde, Denmark; camillaa@ruc.dk (C.A.S.); amandcl@ruc.dk (A.C.); cvbs@ruc.dk (C.S.); malte.mose.tetens.01@regionh.dk (M.M.T.)
* Correspondence: karenak@ruc.dk

**Abstract:** Coronavirus Disease 2019 (COVID-19) has spread across the world, caused lockdowns, and has had serious economic and social consequences. COVID-19 manifests differently in children than adults, as children usually have a milder course of disease, mild symptoms if any, and lower fatality rates are recorded among children. SARS-CoV-2 transmission also seems to be different between children and adults. Many factors are proposed to explain the milder outcome in children, e.g., a more appropriate immune response (especially active innate response), trained immunity, a lack of immunosenescence, and the reduced prevalence of comorbidities. A better understanding of the differences in susceptibility and outcome in children compared with adults could lead to greater knowledge of risk factors for complicated COVID-19 cases and potential treatment targets. We highlight proposed reasons as to why children are less affected by COVID-19 than adults.

**Keywords:** SARS-CoV-2; COVID-19; coronavirus; children; adolescents; young people; ACE2; immune system; transmission; viral infection

## 1. Introduction

In December 2019, the World Health Organization (WHO) picked up on a cluster of respiratory disease cases with unknown cause in Wuhan, China [1]. The respiratory disease, later named 'Coronavirus Disease 2019' (COVID-19), quickly spread to other countries and has since developed into a global pandemic. The disease is caused by the RNA virus 'severe acute respiratory syndrome coronavirus 2' (SARS-CoV-2), and it transmits between people mainly by close contact to an infected individual [2].

There are concerns as to whether children with asymptomatic infection could transmit the virus to other parts of the population. Transmission within schools appears to be dependent on the age of the children. Schools with pupils over the age of 15 have higher transmission rates of the virus, while those with children under the age of 14 have less transmission in the school environment. It is also possible for the virus to be transmitted in a school environment if there is a high rate of community transmission as well [3,4]. Clinical evidence shows that transmission is correlated to symptom severity, hence why asymptomatic children are less likely to transmit the virus [5]. A study showed that almost half of children with detected SARS-CoV-2 were asymptomatic, and the true proportion of asymptomatic children remains unknown [6]. Most of the studies about transmission amongst children are small or were conducted in communities with relatively low levels of transmission. Therefore, the chain of transmission is still unclear, though adults are the main drivers of transmission in society and index cases in families [7].

A common risk factor for admission to an intensive care unit is comorbidities, of which the most important are chronic lung disease, cancer, congenital heart disease, or neurological disease [8]. Though the risk of admission to an intensive care unit is increased due to pediatric comorbidities, it is generally still low for children compared with adults [8]. Age itself also appears to be a risk factor for a more severe course of disease. When children

are categorized by age, there appears to be a lower susceptibility in children (<10–12 years) than adolescents (>10–12 years) when compared to adults [9], suggesting that younger children have a lower risk of contracting the disease. However, young infants under the age of one were reported to be more frequently admitted to the intensive care unit [10].

A serious complication when discussing COVID-19 in children is the development of multisystem inflammatory syndrome in children (MIS-C). MIS-C usually presents after COVID-19 infection and therefore differs from severe COVID-19. Overall, the cases of MIS-C are rare, and although no link is made to comorbidities, pre-existing conditions such as asthma, obesity, and diabetes have been reported [11–14].

We now know that children are less affected by COVID-19 than adults, and serious complications are rare. Children are less involved in the chain of transmission. The impact of the pandemic on the everyday lives of children makes it pivotal that we obtain a better understanding of why children appear to have a better response toward the disease, especially so this can support future decision making in the health care systems, as well as governments worldwide. We aim to highlight the proposed reasons as to why children are less affected by COVID-19 than adults and discuss how these findings should be taken into consideration when debating the consequences of decisions made in the future, such as lockdowns, isolation, masks, and vaccination.

## 2. Factors Contributing to Children Avoiding Severe COVID-19

Overall, there are many suggestions as to why children are less affected by COVID-19 compared to adults. Here, we present and discuss the most promising factors proposed to be involved in pediatric COVID-19 (Figure 1).

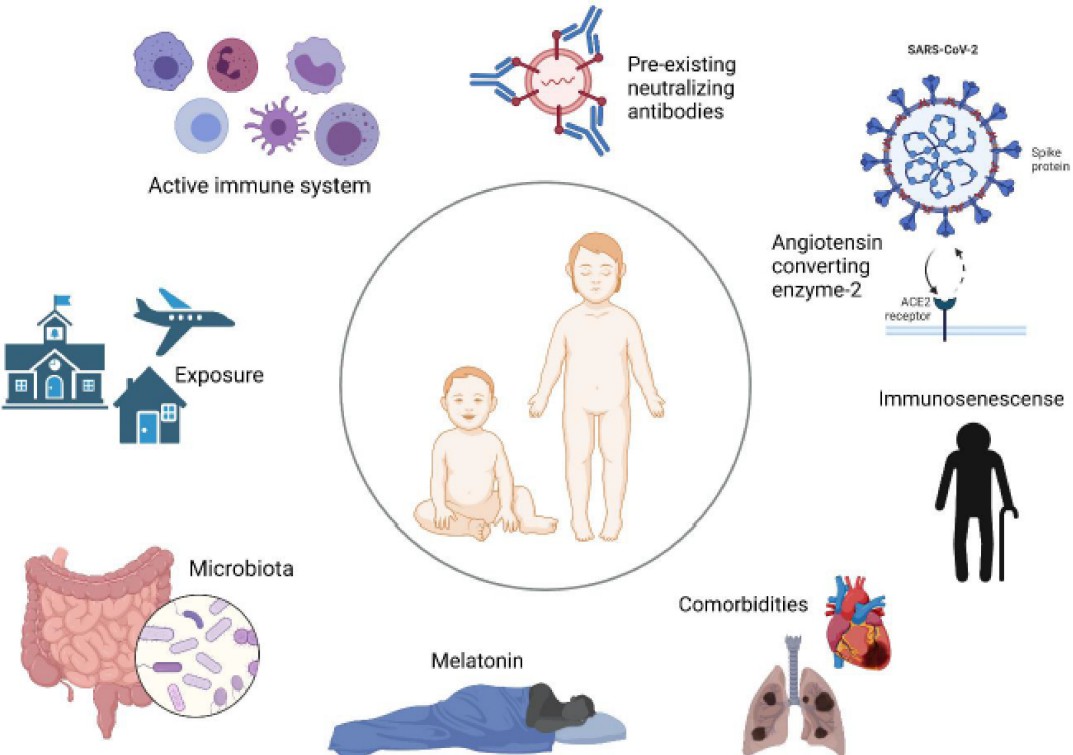

**Figure 1.** Factors proposed to protect against severe pediatric COVID-19.

### 2.1. Strong Innate Response

Children are frequently introduced to viral infections and vaccinations, which keeps their innate immune system active. An increased activation of neutrophils and reduced circulation of various innate immune cells (monocytes, dendritic cells, and natural killer cells) was observed in children with mild SARS-CoV-2 infection in a study, suggesting

that innate immune cells might be recruited to infected areas [15]. In SARS-CoV-2-infected adults only a reduction in circulating non-classical monocytes was observed [15]. This could indicate that children's innate immune response is generally much stronger and efficient at acting against SARS-CoV-2. Specifically, the mucosal immune response appears more vigorous in children compared to adults upon SARS-CoV-2 infection [16]. It has also been reported that the production of IL-17A and IFN-γ during an early immune response can result in a more rapid clearance of viral infection [17].

Children's adaptive immune systems seem to differ from adults', which may partly explain why children's immune systems are less likely to overreact to infection with SARS-CoV-2. Adults show a stronger T-cell response to the spike protein of SARS-CoV-2 and a higher neutralizing antibody count compared to children. However, the boosted adaptive response in adults does not result in better outcomes [17]. A less vigorous adaptive immune response in children and a strong innate response may prevent a cytokine storm, potentially eliminating an unwanted overreaction to the virus.

## 2.2. Activated Immunity Providing Cross-Protection

The cells in the innate immune system can be modified by exposure to antigens [18]. The milder disease in children can be linked to trained immunity offering protection against SARS-CoV-2 by cross-reactive neutralizing antibodies. These cross-reactive antibodies are thought to be more common in children than adults [4,19]. It is possible to activate cross-protection through Bacille Calmette Guerin (BCG) vaccination [19], and has also been demonstrated that measles-containing vaccines (MCVs) can induce adaptive immunomodulation [20]. Although both vaccines reduce overall mortality and protect against viral infections in general [21–23], more research is needed in the area, and currently, the effects of BCG vaccination on COVID-19 is under investigation [24]. Since a large proportion of children have received a number of different vaccines more recently than adults, it could indicate that vaccines play a part in the observed age-related difference in COVID-19 manifestation. However, one study did not find age-related differences in pre-existing antibodies to other coronaviruses, which suggests that cross-immunity does not play an important role in clinical responses to SARS-CoV-2 [17].

## 2.3. The Role of Angiotensin Converting Enzyme-2

The entry point for SARS-CoV-2 into cells is via the receptor, angiotensin converting enzyme-2 (ACE2). ACE2 is part of the renin–angiotensin–aldosterone system (RAAS), which is important for keeping the homeostasis of cardiovascular and respiratory systems [25]. Under normal circumstances, ACE2 is a negative regulator of the RAAS system, it has anti-inflammatory properties, and is involved in converting angiotensin II (Ang II) to angiotensin 1–7 [26]. The virus is able to enter the cells by attaching its Spike (S)-protein to ACE2, which is located on the apical membranes of respiratory epithelium, endothelial cells of blood vessels and heart epithelium, and on enterocytes in the small intestines [18,27–29]. After the entry of SARS-CoV-2 into the body, the expression of ACE2 is downregulated, which causes an increase in the amount of Ang II [18,26]. Elevated Ang II levels can cause severe lung failure, such as that seen with acute lung injury (ALI) and Acute Respiratory Distress syndrome (ARDS) [25,30] (Figure 2).

A study with mice showed that the loss of expression of ACE2 resulted in enhanced vascular permeability, increased lung edema, neutrophil accumulation, and worsened lung function. In addition, mice treated with catalytically active recombinant ACE2 protein showed improvements in the symptoms of ALI and ARDS [30,31]. These findings suggest that ACE2 has a protective role against ARDS and ALI via the negative regulation of Ang II. A study reported the lower expression of ACE2 in nasal and bronchial tissue of children when compared to adults [32]. Studies have supported the findings that the expression of ACE2 in nasal and lung epithelium increases with age and have suggested that this could be an explanation for the difference in COVID-19 severity observed between children and adults [33,34]. However, in the elderly, ACE2 is suggested to decrease again, primarily

based on findings in animal models [26,35], and this age group is most susceptible to severe SARS-CoV-2 infection. Therefore, even though the virus gains access to cells via ACE2, having less of the receptor is not an efficient sole defense against disease, supporting the important role of an active innate immune system in children.

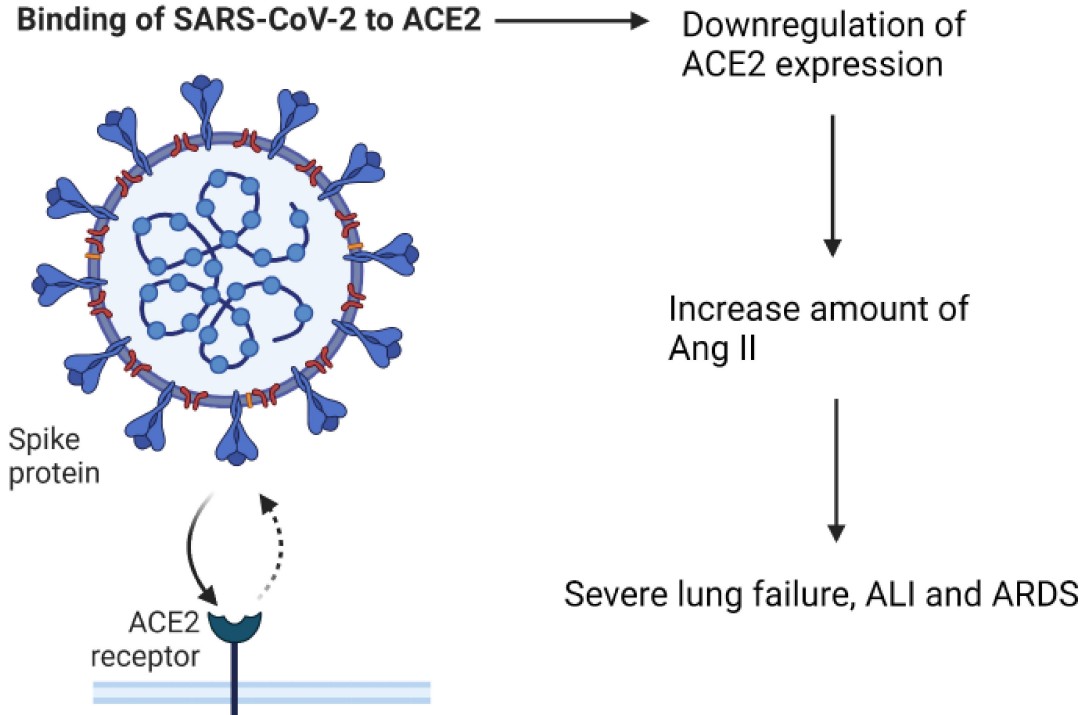

**Figure 2.** The effect of SARS-CoV-2 binding to the ACE2 receptor.

*2.4. Immunosenescence*

Immunosenescence is a term used for the deterioration of the immune system seen with age. It involves changes to both innate and adaptive immune functions, increases the susceptibility to infectious diseases, and decreases the response to vaccinations [36,37]. Since older people seem to be more affected by SARS-CoV-2 infection, it is likely that immunosenescence may contribute to the poor clearance of the virus. Furthermore, the lack of immunosenescence in children could explain why they are less likely to present with a severe infection.

The risk of developing ARDS during a COVID-19 infection is age dependent and can be the cause of death for people infected with SARS-CoV-2 [38]. Adults produce less inflammatory cytokines and antibodies during infection than children due to waning of the immune system [39]. Ageing of the immune system can result in the dysregulation of the production of inflammatory cytokines, which causes damage to the body. Elevated levels of neutrophils and cytokines indicate an exaggerated innate immune response. In adult patients, severe COVID-19 is associated with significantly lower counts of lymphocytes, especially T cells [40,41]. Children have an inherently higher percentage of lymphocytes. Some have suggested that this has a defensive role against SARS-CoV-2. Two studies showed that only 3.5–5.2% of pediatric patients had lymphocytopenia, and the majority had normal leukocytes [42,43].

*2.5. Comorbidities*

A systematic review and meta-analysis from November 2020 considering over 9000 pediatric patients with COVID-19 reported that children with pre-existing conditions had higher risks of severe infection when compared to children with no underlying disease [44]. Pre-existing cardiac conditions were reported in a large proportion of children becoming

critically ill, but also, childhood obesity has been suggested to correlate positively with the severity of COVID-19 [44,45]. However, while comorbidities are associated with severe disease in children, these conditions are not commonly observed. Both adults and children with pre-existing comorbidities have worse outcomes when infected with SARS-CoV-2 than those without any comorbidities, but adults are more likely to suffer from comorbidities [46].

### 2.6. Melatonin Levels

Melatonin is an indoleamine hormone produced at night, mainly by the pineal gland, and secreted into the blood stream. It plays a big role in the biological processes of the circadian system, regulating night–day as well as sleep–wake cycles, and has anti-inflammatory and anti-oxidative properties [47,48]. SARS-CoV-2 is considered to originate from bats; however, it has minimal to no impact on the health of the host. Bats are nocturnal animals and have high levels of melatonin compared to humans, which may contribute to their increased resistance toward illness from the virus [49,50]. Melatonin levels in humans are negatively correlated with age [51,52], potentially contributing to the worse symptoms experienced in the elderly and explaining why children are better protected when infected with SARS-CoV-2. Melatonin has been shown to increase the proliferation of T and B cells, natural killer cells, granulocytes, and monocytes in bone marrow and tissues [53] and can also protect against ARDS [54]. A recently published review highlights several steps where melatonin can interfere with damages caused by COVID-19. Infection by SARS-CoV-2 can result in the high production of neutrophil myeloperoxidase (MOP) and reactive oxygen species (ROS), both involved in combating pathogens, but can cause tissue damage if the response is too strong. Melatonin can inhibit MOP activity, as well as scavenger ROS, thereby potentially reducing COVID-19 severity [55]. Melatonin as a possible supplement against COVID-19 is particularly interesting due to the stability, accessibility, safety, and cost of the drug.

### 2.7. Difference in Microbiota

Another less-examined hypothesis as to why children may fare better against COVID-19 involves the microbiota. The microbiota is important for human well-being; it is involved in regulating the immune system, inflammation, and gut homeostasis and plays an important part in protection against pathogens. The gut microbiome seems to be altered in patients with COVID-19, with an under-representation of *Faecalibacterium prausnitzii, Eubacterium rectale,* and bifidobacteria, which all have immunomodulatory prospects. This change in microbial composition, associated with elevated levels of inflammatory cytokines and blood markers of tissue damage, suggests the involvement of the gut microbiome in COVID-19 disease severity [56]. The gut microbiota changes with age, and children generally have higher numbers of *Bifidobacterium* than adults [57], potentially providing a better defense against infection. Children might also have a higher nasopharyngeal colonization of viruses and bacteria, which by competition might limit the growth of SARS-CoV-2 and lead to the reduced colonization of the pathogen. However, it is important to remember that the microbiota is affected by many factors and that the reported changes might be a cause of the disease and not vice versa.

### 2.8. Exposure to SARS-CoV-2 and Viral Load

Generally, young children have less exposure to SARS-CoV-2 due to their behavior and lifestyle. They follow a more planned schedule, meet the same people, and do not have, e.g., work, travel, and shopping as part of their everyday lives, where different encounters occur. Viral exposure is associated with viral load, which influences the intensity of the disease [58–60]. A study with 76 adult patients from a hospital in Nanchang in China, from the start of 2020, shows that the mean viral load was around 60 times higher for cases with severe COVID-19 compared to cases with mild COVID-19. This suggests that higher viral loads might be associated with severe clinical outcomes. Cases with severe

disease tend to have a longer viral shedding period than those only mildly affected and can therefore transmit the virus for a longer time [60]. These findings may be consistent with the hypothesis that children have less severe and shorter disease periods because of lower viral loads. It was shown that a lower viral load is connected to no or mild symptoms [61]. However, a study has shown that viral load in children is independent of symptoms and disease severity, again supporting the hypothesis that the immune system may play an important role in mild COVID-19 symptoms in children [62].

As previously mentioned, it is possible that the presence of other viruses in the respiratory tract of young children limits the establishment of SARS-CoV-2 by direct virus–virus interactions and/or viral exclusion during colonization. This hypothesis provides a link between the severity of the disease and the size of viral load.

## 3. Discussion

Overall, there are many studies regarding COVID-19 in children; however, the participant number is generally low, and they are from many different parts of the world. These smaller studies make it difficult to fully understand how children are affected and why they are affected differently. Furthermore, we only included published, peer-reviewed studies, so new information from pre-print studies was not considered. Once these are published, they might add to the understanding of how COVID-19 affects children.

The lack of information about children with COVID-19 might be further explained by the way the disease presents in children. Children might be under-represented when it comes to confirmed cases of COVID-19. When compared to adults, children have significantly reduced severity of infection, meaning they are not sick enough to be tested. This lack of symptoms does not meet the case definition for COVID-19, and therefore, children do not qualify for testing in some places, leading to a low number of confirmed cases. If infected children are not identified, it is not possible to include them in studies. However, it could be argued that tests should not be performed in individuals without symptoms, as this is not practice for other infectious diseases.

The immune system naturally changes with age; however, this is a gradual change that takes place throughout a lifetime. The immune system is constantly developing during childhood, which makes it unclear at which age the immunological hypotheses for COVID-19 applies and to what degree. It could seem that young infants and adolescents have higher risks of severe COVID-19 than children from around 1–14 years; however, overall, children have better clinical outcomes than adults. Elevated inflammatory markers in children are less common, and normally, healthy children produce less cytokines than healthy adults [63], which could be one of the reasons why they experience a milder disease course. It is also important to remember that children can also be susceptible to other respiratory viral infections, but usually with quick recovery.

There is focus on antibodies, vaccines, and the overall strengthening of adaptive responses, which in research terms are also easier to measure. A lot of research has gone into the development of different SARS-CoV-2 vaccines, which is relevant for the weak and immunocompromised and might play a part in reducing the transmission of COVID-19 in society. However, vaccination against COVID-19 in previously infected individuals and overall healthy individuals, especially children, is a subject of great debate. Focusing on vaccines and the adaptive immune response might be taking focus from the innate and cellular immune response, which is highly active in children and functions as a first line of defense. In children, the immune system is under development and is constantly being introduced to new pathogens. Children present with a higher number of innate immune cells in their upper respiratory tract compared to adults [64], and nasal immune cells are also observed to decrease with age [65]. In relation to COVID-19, innate defense may play a crucial role in disease severity and explain why children present with better outcomes during infection. The overall lack of research reaching beyond adaptive responses are scarce, and in particular, a focus on immune defenses such as mucosal immunity, antimicrobial peptides, the pathogen recognizing Toll-like receptors, and the opsonizing

complement pathway, including the specific invertebrate lectin pathway, would be relevant moving forward.

COVID-19 vaccination strategies in children should be given careful thought. Until now, other vaccines have been given to specific compromised groups of people or the whole population to prevent serious invalidating and lethal diseases. One should consider whether children are at risk due to SARS-CoV-2 and whether children are the spreaders of the disease to decide upon prevention strategies. When considering the implications that have followed COVID-19 (lockdowns, isolation, testing, etc.), it is important to put these in perspective to the danger of the disease. In children, most studies have shown mild symptoms, if any, and fatality is low. Keeping this in mind, implications such as school and sports cancellations, regular testing, and possible isolation might have a greater impact on children than the actual disease, not just mentally but also physically. The immune response in children is, like the child itself, developing and learning. However, by keeping children at home, the lack of normal introduction to various microbes might lead to an under-stimulated immune system, which could cause complications. The hygiene hypothesis was first introduced in 1989 [66], and although it has been criticized for focusing too much on specific hygiene, such as handwashing, it is still undeniable that the immune system needs exposure to develop. A lack of exposure to pathogens might lead to allergies or autoimmune deficiencies later. This also brings in another question to consider: will COVID-19 become a childhood disease? Since children are less affected by COVID-19, it has been proposed that the disease will become accepted as a mild childhood disease, providing natural immunity and protection later in life, where the disease appears to be more severe.

The possible explanations for why children are affected differently by COVID-19 than adults are many, and most likely, it is a combination of factors. Immunological factors could include a more active innate immune response, cross-protection from trained immunity, and the lack of immunosenescence. Other factors could be the expression and function of ACE2, viral load, various environmental and lifestyle factors, as well as less comorbidities when comparing to adults and especially the elderly. Studies so far have focused on retrospective prevalence. Future studies should be designed to follow individuals prospectively to assess the degree and mode of viral transmission in families, to identify index cases in families, and elucidate the role of children during the COVID -19 pandemic. Future prospective studies could help provide better understanding of why children have better outcomes than adults during and after infection.

**Funding:** This research was funded by the Lundbeck Foundation, grant number R349-2020-703.

**Acknowledgments:** Karen A. Krogfelt is a member of the Pandemix centre at Roskilde University.

**Conflicts of Interest:** The authors declare no conflict of interest.

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
