# Peer review of "Children Naturally Evading COVID-19—Why Children Differ from Adults"

_covid, doi:10.3390/covid2030025_

Round 1

Reviewer 1 Report

Majorr Comments:

  1. Lines 47-49: This sentence needs to be revised or deleted.  In the previous sentence the authors state that “adults are the main drivers of transmission in society and index cases in families.”  This wording conflicts with the idea of children as “carriers.”  If adults are truly the index cases in families, how did the children become carriers prior to the infection of adults?  The word “carriers” does not appear in reference 7.
  2. Lines 50-52: This sentence appears to list general risk factors for severe COVID and then notes that children often do not suffer from these comorbidities.  The sentence should be revised to make clear that the authors are referring to pediatric comorbidities and still make the point that these comorbidities are relatively rare compared to the adult population. 
  3. Lines 103-103: “The milder disease in children can be linked to trained immunity that might offer cross protection against SARS-CoV-2 by neutralizing cross-reactive antibodies.”  I’m not sure but I don’t think this sentence is what the authors intended to write.  I think the “cross-reactive antibodies” is supposed to be “virus.” 
  4. Delete section 3.2. Line 105-107:  “It is shown that it is possible to activate cross-protection…”  This sentence is unclear and needs to be re-worded.  The current sentence is factually incorrect.  Reference 20 refers to measles immune suppression following infection, NOT vaccination.  Ref 20 is about “the immunosuppressive effects of measles to depletion of B and T lymphocytes” and provides “an explanation for the long-term BENEFITS of measles vaccination in preventing all-cause infectious disease.”  Bearing this in mind, the effects of the measles vaccine has nothing to do with eliciting cross-protective immunity.  And since BCG appears to have no effects on COVID (Ref 24), this sub-section should be deleted.
  5. Lines 155-156:  “Adults produce less inflammatory cytokines during infection than children due to the weaning of the immune system.”  Do you mean “waning”?  If so, please provide references.  In animal studies, aged animals can produce higher levels of inflammatory cytokines than young animals.  Immunosenescence isn’t simply a reduction in the ability of the immune system to response, but a reduced ability to regulate immune responses, either positively or negatively.  Lines 155-156 seem to contradict lines 156-158. 
  6. Lines 269-270: “The innate defense in children is non-specific, compared to the adult innate defense.”  Please provide a reference to support this statement.  Innate immune receptors are the same in children and adults.  TLR3 when you are three years old is still TLR3 when you are thirty years old.  Beta-coronaviruses other than SARS-CoV-2 are in circulation.  SARS-CoV-2 is not a different class of pathogen than OC43.
  7.  

Minor Comments:

  1. Lines 61-62: Correct spelling “MIS-Care rare” to “MIS-C are rare”
  2. Line 63 – I think there is an extra space between “is” and “made”
  3. Line 75 – Section #2 is missing. The introduction is section #1 and leads directly into section #3 on line 75.  Renumber sections.
  4. Lines 97-99: “A less vigorous adaptive…overreaction to the virus.”  I don’t think this is a complete sentence.
  5. Line 141: Correct spelling – “anactive”
  6. Line 162: Strike “Chinese.”  The nationality of the researchers can be determined by checking the reference info.
  7. Lines 167-170: Is reference 43 still the best reference to support this statement.  Have more recent studies on this topic been published?  If so, please revise and include.

Other comments (hopefully helpful):

  1. The quality of the English is sufficient, but the writing could be improved throughout. For example, line 66 could be revised to read “Children appear less involved in the chain…”  Lines 67-68 could be revised to read “The impact of the pandemic on the everday lives of children makes it pivotal that we obtain…”  Generally you should avoid writing “there is” or “there are” (expletive constructs) and the word “this” should never be followed by a verb.  Similarly, “it” should be avoided as the subject of a sentence unless the wording is very clear about what “it” is.  For example, in lines 90-92, “It has also been reported that” could be deleted without changing the meaning of the sentence.  In line 226 “a study has shown that” could be deleted without changing the meaning of the sentence.

Reviewer 2 Report

This is a very interesting manuscript, and  the text is well- written.

Round 2

Reviewer 1 Report

The authors have addressed most of my points.  However, the manuscript is still not suitable for publication.  

Please correct the following problems.

  1. Lines 106-108.  Authors refer to "immune suppression by measels-containing vaccines (MCV) [19]."  Reference 19 deals with immunosuppression from measles infection and NOT from measles vaccination.  This portion of the manuscript must be corrected as the current version is a gross misuse of this reference.  I'm not aware of any measles vaccine-mediated immunosuppression and the authors do not provide any references supporting the existence of such.  Given the strong anti-vaccine sentiment in much of society today, it's very important that the authors' statements on this point be factually correct.  

2.   Lines 156-158.  The comparison between varying levels of antibody in adult and pediatric populations is not supported by this reference.  Reference 38 in the revised manuscript examines the decline in SARS-CoV-2 vaccine specific antibodies in health care workers six months following immunization.  The study does not include a pediatric group.

3.  Lines 269-272.  "The innate immune system in children has a broader range, compare to adults [63]."  This sentence was revised by the authors, but they still do not address my points.  What do they mean by "range"?  Reference 63 states that "TLR7/8 and TLR9 driven IFN-a2 responses increased to reach adult levels by 1 year of life."    The reference also states that "RIG-induced IFN-a production in PBMCs from human neonates and young children aged 12-59 months was found significantly lower than in adults" and "thus indicated that in addition to TLR-, other PRR-mediated responses are also attenuated in early life."  There are clearly age-related effects on the innate response, but the authors use of this reference to support the idea that the innate response in children is more effective against a wide range of pathogens is not correct.  As with my point above, the reference used by the authors is not appropriate.  

The use of correct references is particularly important in a review article.  If the authors cannot find references to support their statements, then they should change their statements.  The weaknesses in the revised manuscripts were weaknesses in the original manuscript that the authors did not sufficiently address.  
